# A Novel Adaptive Sensor Fault Estimation Algorithm in Robust Fault Diagnosis

**DOI:** 10.3390/s22249638

**Published:** 2022-12-08

**Authors:** Marcin Pazera, Marcin Witczak

**Affiliations:** Institute of Control and Computation Engineering, University of Zielona Góra, ul. Prof. Z. Szafrana 2, 65-516 Zielona Góra, Poland

**Keywords:** fault estimation algorithm, sensor fault estimation, fault estimation, robust fault diagnosis

## Abstract

The paper deals with a robust sensor fault estimation by proposing a novel algorithm capable of reconstructing faults occurring in the system. The provided approach relies on calculating the fault estimation adaptively in every discrete time instance. The approach is developed for the systems influenced by unknown measurement and process disturbance. Such an issue has been handled with applying the commonly known $H_{\infty }$ approach. The novelty of the proposed algorithm consists of eliminating a difference between consecutive samples of the fault in an estimation error. This results in a easier way of designing the robust estimator by simplification of the linear matrix inequalities. The final part of the paper is devoted to an illustrative example with implementation to a laboratory two-rotor aerodynamical system.

## 1. Introduction

In the developing world, the expansion toward Industry 4.0 results in an increase in the components of the system. Any industrial system cannot be imagined without measuring devices. Especially these days, measured information is analyzed in many ways to achieve an optimal performance of the entire technological process. Thereby, industries increase the number of sensors to have the information of the process as accurate as possible. However, by increasing the pool of sensors, it increases a chance of fault occurrence in some of them.

The developments concerning fault diagnosis (FD) have received a significant scientific attention over the last decades, and a large pool of reliable FD strategies is available [1,2,3,4]. However, while screening these fundamental works on FD, one can observe that initially, the research was focused on fault detection and isolation (FDI). This situation was completely changed along with the number of works devoted toward fault-tolerant control (FTC) [5,6,7]. Indeed, fault estimation or identification constitutes the crucial element of all active FTC schemes FTC [8,9]. This simply means that the FTC performance depends on the knowledge about the faults, which is provided along with the fault estimation. There are several approaches devoted to either sensor or actuator fault estimation [10,11,12]. The main development trends of fault estimation are oriented toward dedicated observer-based approaches [13,14,15]. Their appealing property is that they can realize both FDI and fault estimation simultaneously. Note also that the problem of simultaneous sensor and actuator estimation has received considerable attention as well [16,17,18,19,20]. However, the design strategies are usually realized by a simple extension of the actuator/sensor fault estimation schemes. Moreover, a recent literature review clearly indicated the trends for settling fault estimation for nonlinear systems. Indeed, in [21,22,23], the authors transformed a nonlinear Lipschitz system into a linear parameter-varying (LPV) one with the so-called reoriented Lipschitz strategy. Such an approach enables optimal $H_{\infty }$ fault estimates in the limited frequency range. An alternative approach to Lipschitz systems was described in [24]. In the proposed strategy, the system is split into two subsystems. Each subsystem is affected by either an actuator or a sensor fault. This results in a tandem of separated sliding mode observers (SMOs). The main issue of the approach is that the resulting fault estimation error converges asymptotically to zero without taking inevitable disturbances into account. This unappealing property is eliminated in [25,26] by taking into account bounded disturbances. As an alternative to Lipschitz strategies, one can use LPV or Takagi–Sugeno fuzzy ones. A representative example of such strategies is provided in [27,28]. This paper proposes the so-called adaptive fuzzy estimator while considering predefined fault scenarios: drift, bias, and loss of accuracy along with loss of effectiveness. The main drawback of this approach is that it is not robust to disturbances, which may significantly impair its performance. Another interesting Takagi–Sugeno approach is proposed in [29] for switching nonlinearities. However, it inherits the same drawback with respect to the lack of robustness. Subsequently, the work [30] proposes a new Takagi–Sugeno multiple integral unknown input observer, which decouples disturbances. The final group of strategies deals with polynomial LPV and Takagi–Sugeno systems [31,32]. The approach reduces to converting the system into the so-called augmented-state form. To summarize, the dominating number of fault estimators is based on the following general frameworks:Adaptive estimators [33,34].High-gain and sliding-mode observers [26,35,36].Kalman filter-based [37,38].Virtual diagnostic estimators [39].Proportional–integral (PI) observers [16].

Nevertheless, a typical way of obtaining the sensor faults estimates is that it is developed with a set of observers [40,41]. Each observer uses all but one sensor, and estimates the missing sensor readings. Those estimates are then compared with real sensor measurements and, as a result, the sensor fault is obtained. An obvious drawback of this scheme is that it is assumed that, within a given time interval, only one sensor reading is impaired by a fault. The other approaches of course allow for estimating all sensor faults at the same time, although, the unwelcome rate of change of the sensor fault factor is to be additionally minimized, which also influences the design involving such an approach, due to the fact that this additional factor needs to be taken into account during the optimization process while designing the gain matrices. To sort out such an issue, a novel structure of the observer is proposed. It merges two previous approaches proposed by the authors, namely the direct estimation strategy, where the sensor fault is estimated based on the output equation with the adaptive approach. However, the proposed algorithm minimizes the rate of change of the fault factor, which is its main advantage. Additionally, the investigated approach is capable of handling process and measurement uncertainties by an application of $H_{\infty }$ theory.

The paper is organized as follows: Section 2 introduces the problem and provides a set of necessary preliminaries. In Section 3, a novel sensor fault estimation scheme is proposed along with the stability analysis. Section 4 provides an illustrative example dealing with a laboratory multi-tank system. Finally, Section 5 concludes the paper.

## 2. Preliminaries

Let us start by defining the possibly faulty system given by
(1)$$x_{k+1}=Ax_{k}+Bu_{k}+h(x_{k})+W_{1}w_{1,k},$$
(2)$$y_{k}=Cx_{k}+ff_{k}+W_{2}w_{2,k}.$$
where *k* stands for a discrete time instance and $x_{k}=\left[x_{1},x_{2},\ldots ,x_{n}\right]\in R^{n}$, $u_{k}=\left[u_{1},u_{2},\ldots ,u_{r}\right]\in R^{r}$, $y_{k}=\left[y_{1},y_{2},\ldots ,x_{m}\right]\in R^{m}$, stand for the state, control input and measured output vectors, respectively. Moreover, $f_{k}=\left[f_{1},f_{2},\ldots ,f_{n_{y}}\right]\in R^{n_{y}}$ is the fault vector which affects the measured output, and for such a reason it will be referred to as a *sensor fault*. Moreover, $w_{1,k}=\left[w_{1,1},w_{1,2},\ldots ,w_{1,q_{1}}\right]\in R^{q_{1}}$ and $w_{2,k}=\left[w_{2,1},w_{2,2},\ldots ,w_{2,q_{2}}\right]\in R^{q_{2}}$ are unknown exogenous process measurement uncertainty vectors, respectively. Note that both $w_{1,k}$ as well as $w_{2,k}$ belong to $l_{2}$ class, and hence, they obey
(3)$$\begin{array}{cc} l_{2} & =\{w\in R^{n}|\;\;{∥w∥}_{l_{2}}<+\infty \}, \end{array}$$
(4)$$\begin{array}{cc} {∥w∥}_{l_{2}} & ={\left(\sum_{k=0}^{\infty }{∥w_{k}∥}^{2}\right)}^{\frac{1}{2}}, \end{array}$$
meaning that they have a finite energy. Moreover, a matrix *f* denotes a sensor fault distribution one. In the other words, it describes the way the sensor fault vector influences the system measurements. Furthermore, $h(x_{k}):X\to X$ represents a non-linear function with respect to the state. It is assumed that
$h(x_{k})$ is of the Lipschitz form, i.e.,
(5)$$\|h(x)-h(y)\|\leq \gamma_{g}\left\|x-y\right\|,\;\forall x,y\in X.$$

Evidently, a way of dealing with such Lipschitz nonlinearities is described, e.g., in [42,43].

The problem stated in this paper is related to reconstructing sensor faults. The sensor faults are as important as the other ones, such as, for example, actuator faults, due to the fact that it is hard for any system without measurements. Of course, there are plenty of effective ways and approaches capable of estimating it; however, adaptive observers presented in the literature (see [42,44,45]) are burdened with an unwelcome rate of change of the sensor fault factor. Approaching the problem in such a way determines the minimization of that additional factor in the design process, which additionally might worsen the final quality of estimation and as a consequence, a quality of control and final product. The issue stated in this paper concerns the problem of estimating the state and sensor fault for which the rate of change factor is eliminated by developing a suitable structure of the observer. In the subsequent section, an observer design procedure for handling such a problem is provided.

## 3. Fault Estimation

To settle an issue formulated in the former section, let us propose an observer of the following structure: (6)$${\overset{\wedge }{x}}_{k+1}=A{\overset{\wedge }{x}}_{k}+Bu_{k}+h({\overset{\wedge }{x}}_{k})+K_{x}(y_{k}-{\overset{\wedge }{y}}_{k}),$$(7)$${\overset{\wedge }{f}}_{k}=Zy_{k}-\mathrm{ZCA}{\overset{\wedge }{x}}_{k-1}-\mathrm{ZCB}u_{k-1}-\mathrm{ZCh}({\overset{\wedge }{x}}_{k-1})+K_{s}(y_{k-1}-{\overset{\wedge }{y}}_{k-1}),$$(8)$$\begin{array}{cc} {\overset{\wedge }{y}}_{k} & =C{\overset{\wedge }{x}}_{k}+f{\overset{\wedge }{f}}_{k}, \end{array}$$
where ***K****x* and ***K****s* are designated state and sensor fault estimation matrices. Moreover, ***Z*** stands for a pseudo-inversion of the sensor fault distribution matrix *f* such that $Z={\left(f\right)}^{†},$Zf=I. The structure of the observer stands for a combination of an estimator enabling the direct fault estimation [46] with the one able to estimate the fault adaptively [42]. Such an algorithm inherits the advantages of both these both approaches.

To realize the design procedure, let us start with calculating the state estimation error as
(9)$$\begin{array}{cc} e_{k+1} & =\left[A-K_{x}C\right]e_{k}+h(x_{k})-h({\overset{\wedge }{x}}_{k})-K_{x}fe_{s,k}+W_{1}w_{1,k}-K_{x}W_{2}w_{2,k}, \end{array}$$
with $e_{s,k}$ indicating a sensor fault estimation error. Before proceeding to the sensor fault estimation error, let us transform the output Equation (Equation 2) in such a way as to obtain the sensor fault formula: (10)$$f_{k}=Z\left(y_{k}-Cx_{k}-W_{2}w_{2,k}\right),$$
and even more specifically
(11)$$f_{k}=Z(y_{k}-\mathrm{CA}x_{k-1}-\mathrm{CB}u_{k-1}-\mathrm{Ch}(x_{k-1})-CW_{1}w_{1,k-1}-W_{2}w_{2,k}).$$

Thereby, taking into account the sensor fault (Equation 11) and its estimate (Equation 7), the dynamics of the sensor fault estimation error can be presented as
(12)$$\begin{array}{cc} e_{s,k} & =f_{k}-{\overset{\wedge }{f}}_{k}=Zy_{k}-\mathrm{ZCA}x_{k-1}-\mathrm{ZCB}u_{k-1}-\mathrm{ZCh}(x_{k-1})-\mathrm{ZC}W_{1}w_{1,k-1}\\ \; & -{\mathrm{ZW}}_{2}w_{2,k}-Zy_{k}+\mathrm{ZCA}{\overset{\wedge }{x}}_{k-1}+\mathrm{ZCB}u_{k-1}+\mathrm{ZCh}({\overset{\wedge }{x}}_{k-1})-K_{s}Cx_{k-1}\\ \; & -K_{s}ff_{k-1}-K_{s}W_{2}w_{2,k-1}+K_{s}C{\overset{\wedge }{x}}_{k-1}+K_{s}f{\overset{\wedge }{f}}_{k-1}\\ \; & =\left[-\mathrm{ZCA}-K_{s}C\right]e_{,k-1}-\mathrm{ZCh}(x_{k-1})+\mathrm{ZCh}({\overset{\wedge }{x}}_{k-1})\\ \; & -K_{s}fe_{s,k-1}-K_{s}W_{2}w_{2,k-1}-{\mathrm{ZCW}}_{1}w_{1,k-1}-{\mathrm{ZW}}_{2}w_{2,k}. \end{array}$$

Then, for further analysis, let the sensor fault estimation error be shifted in time by one sample, resulting in
(13)$$\begin{array}{cc} e_{s,k+1} & =\left[-\mathrm{ZCA}-K_{s}C\right]e_{,k}-\mathrm{ZCh}(x_{k})+\mathrm{ZCh}({\overset{\wedge }{x}}_{k})\\ \; & -K_{s}e_{s,k}-K_{s}W_{2}w_{2,k}-{\mathrm{ZCW}}_{1}w_{1,k}-{\mathrm{ZW}}_{2}w_{2,k+1}, \end{array}$$
which is actually equal to (Equation 8), an evolution of the sensor fault with the signal at time k+1. Having such estimation errors for both state and sensor faults, let us compose a vector incorporating both of them. However, before doing this, let us introduce the following lemma:

**Lemma** **1**([47]). *For **h***(*·*)*, the following statements are equivalent:*
 *1.****h**(·) is Lipschitz with respect to their arguments with Lipschitz constant $\gamma_{g}>0$, i.e.,*$$\left\|h(X)-h(Y)\right\|\leq \gamma_{g}\left\|X-Y\right\|,\;\forall X,Y\in X.$$ *2.**For all i,j=1,…,n, there exist functions $g_{i,j}:X\times X\overset{}{\to }R$ and constants ${\underline{\gamma }}_{g_{i,j}}$ and ${\bar{\gamma }}_{g_{i,j}}$ such that, for each X,Y∈X*(14)$$h(X)-h(Y)=\sum_{i=1}^{n}\sum_{j=1}^{n}g_{i,j}G_{i,j}(X-Y),$$*and*(15)$${\underline{\gamma }}_{g,i,j}\leq g_{i,j}\leq {\bar{\gamma }}_{g,i,j},$$(16)$$g_{i,j}≜g_{i,j}\left(X^{Y_{j-1}},X^{Y_{j}}\right),\;G_{i,j}=c_{i}c_{i}^{T},$$*as well as a scalar function $g_{i,j}$ given by*(17)$$\begin{array}{cc} g_{ij} & \left(X^{Y_{j-1}},X^{Y_{j}}\right)=\left\{\begin{array}{cc} 0 & \mathrm{if}\;x_{j}=y_{j}\\ \frac{h_{i}\left(X^{Y_{j-1}}\right)-h_{i}\left(X^{Y_{j}}\right)}{x_{j}-y_{j}} & \mathrm{if}\;x_{j}\neq y_{j}, \end{array}\right. \end{array}$$*where **c**_i_ stands for the i-th column of the n-th order identity matrix, while $X^{Y_{i}}$ is defined by*(18)$$\begin{array}{cc} X^{Y_{i}} & =\left[\begin{array}{c} y_{1}\\ \vdots \\ y_{i}\\ x_{i+1}\\ \vdots \\ x_{n} \end{array}\right],\;\;\mathrm{for}\;\;i=1,\ldots ,n \end{array}$$(19)$$\begin{array}{cc} X^{Y_{0}} & =X. \end{array}$$

Thus, the compact error vector takes the form
(20)$${\tilde{e}}_{k+1}=\tilde{X}(g){\tilde{e}}_{k}+\tilde{Z}{\tilde{d}}_{k},$$
where
(21)$$\begin{array}{cc} {\tilde{e}}_{k} & =\left[\begin{array}{c} e_{k}\\ e_{s,k} \end{array}\right] \end{array}$$
(22)$$\begin{array}{cc} {\tilde{d}}_{k} & =\left[\begin{array}{c} w_{1,k}\\ w_{2,k}\\ w_{2,k+1} \end{array}\right] \end{array}$$
while $\tilde{X}\left(g\right)=\tilde{A}\left(g\right)-\tilde{K}\tilde{C}$ and $\tilde{Z}={\tilde{W}}_{1}-\tilde{K}{\tilde{W}}_{2}$ with
(23)$$\begin{array}{ccc} \tilde{A}\left(g\right)=\left[\begin{array}{cc} A+A\left(g\right) & 0\\ \mathrm{TA}+\mathrm{TA}\left(g\right)T^{T} & 0 \end{array}\right], & \tilde{K}=\left[\begin{array}{c} K_{x}\\ K_{s} \end{array}\right], & \tilde{C}=\left[\begin{array}{c} C\;f \end{array}\right],\\ {\tilde{W}}_{1}=\left[\begin{array}{ccc} W_{1} & 0 & 0\\ {\mathrm{TW}}_{1} & 0 & -{\mathrm{ZW}}_{2} \end{array}\right], & {\tilde{W}}_{2}=\left[\begin{array}{ccc} 0 & W_{2} & 0 \end{array}\right], & T=-\mathrm{ZC}. \end{array}$$

In this point, it can be easily noticed the difference of the proposed approach compared to the classical adaptive observer. The vector ${\tilde{d}}_{k}$ in (Equation 22) does not contain the rate of change of the fault, unlike the strategy proposed in ref. [42]. This simply shows the advantages of such an approach due to the fact that the unwelcome rate factor does not need to be taken into account in the optimization process.

Having in mind the above deliberations, an $H_{\infty }$ observer design procedure can be handled by shaping the following theorem:

**Theorem 1.** 
*Assume that the nonlinear system satisfies condition 1 of Lemma 1. Thus, it can be approximated by *(*Equation 20*)*. Then the design problem of the observer *(*Equation 6*)*–*(*Equation 8*)* for the system given by *(*Equation 1*)*–*(*Equation 2*)* is solvable for a prescribed attenuation level μ>0 of the state and estimation error *(*Equation 20*)*, if there exist matrices P≻0 and **N** such that the following condition is met:*

(24)
$$\left[\begin{array}{ccc} I-P & 0 & {\tilde{A}}^{T}(g)P-{\tilde{C}}^{T}N^{T}\\ 0 & -\mu^{2}I & {\tilde{W}}_{1}^{T}P-{\tilde{W}}_{2}^{T}N^{T}\\ P\tilde{A}\left(g\right)-N\tilde{C} & P{\tilde{W}}_{1}-N{\tilde{W}}_{2} & -P \end{array}\right]≺0.$$



**Proof.** The problem of designing the $H_{\infty }$ estimator reduces to find the ***N***, ***U*** and ***P*** matrices by solving (Equation 24) such that
(25)$$\begin{array}{cc} \lim_{k\to \infty }{\tilde{e}}_{k}=0\; & \mathrm{for}\;{\tilde{d}}_{k}=0, \end{array}$$
(26)$$\begin{array}{cc} \|{\tilde{e}}_{k}{\|}_{l_{2}}<\mu \|{\tilde{d}}_{k}\|l_{2}\; & \mathrm{for}\;{\tilde{d}}_{k}\neq 0,{\tilde{e}}_{0}=0. \end{array}$$
It is known that for the sake of the $H_{\infty }$ design, a Lyapunov function is equivalent to
(27)$$\Delta V_{k}+{\tilde{e}}_{k}^{T}{\tilde{e}}_{k}-\mu^{2}d_{k}^{T}d_{k}<0,$$
where: $\Delta V_{k}=V_{k+1}-V_{k}$, $V_{k}={\tilde{e}}_{k}^{T}P{\tilde{e}}_{k}$ and P≻0, which is sufficient to solve that problem. It is evident that for ${\tilde{d}}_{k}=0$, inequality (Equation 27) boils down to
(28)$$\Delta V_{k}+{\tilde{e}}_{k}^{T}{\tilde{e}}_{k}<0,$$
with $\Delta V_{k}=V_{k+1}-V_{k}$, and then leads to (Equation 25). Thus, by employing (Equation 20), it is easy to show that
(29)$$\begin{array}{cc} \Delta V_{k}+P{\tilde{e}}_{i,k}^{T}{\tilde{e}}_{i,k}-\mu^{2}d_{k}^{T}d_{k} & ={\tilde{e}}_{k}^{T}\left(\tilde{X}{\left(g\right)}^{T}P\tilde{X}\left(g\right)+I-P\right){\tilde{e}}_{k}\\ \; & +{\tilde{e}}_{k}^{T}\left(\tilde{X}{\left(g\right)}^{T}P\tilde{Z}\right){\tilde{d}}_{k}+{\tilde{d}}_{k}^{T}({\tilde{Z}}^{T}P\tilde{X}\left(g\right)){\tilde{e}}_{k}\\ \; & +{\tilde{d}}_{k}^{T}({\tilde{Z}}^{T}P\tilde{Z}-\mu^{2}I){\tilde{d}}_{k}≺0. \end{array}$$
Then, by establishing a new temporary variable
(30)$$w_{k}=\left[\begin{array}{c} {\tilde{e}}_{k}\\ {\tilde{d}}_{k} \end{array}\right],$$
inequality (Equation 29) may be reconstructed into the following shape:
(31)$$w_{k}^{T}\left[\begin{array}{cc} \tilde{X}{\left(g\right)}^{T}P\tilde{X}\left(g\right)-P+I & \tilde{X}{\left(g\right)}^{T}P\tilde{Z}\\ {\tilde{Z}}^{T}P\tilde{X}\left(g\right) & {\tilde{Z}}^{T}P\tilde{Z}-\mu^{2}I \end{array}\right]w_{k}≺0,$$
or alternatively into another form:
(32)$$\left[\begin{array}{c} \tilde{X}{\left(g\right)}^{T}\\ {\tilde{Z}}^{T} \end{array}\right]P\left[\begin{array}{c} \tilde{X}\left(g\right)\;\tilde{Z} \end{array}\right]+\left[\begin{array}{cc} I-P & 0\\ 0 & -\mu^{2}I \end{array}\right]≺0.$$
Afterwards, it leads to
(33)$$\left[\begin{array}{ccc} -P+I & 0 & \tilde{X}{\left(g\right)}^{T}P\\ 0 & -\mu^{2}I & {\tilde{Z}}^{T}P\\ P\tilde{X}\left(g\right) & P\tilde{Z} & -P \end{array}\right]≺0,$$
by applying the Schur complement to (Equation 32) with left- and right-side multiplication by $\mathrm{diag}\left(I,I,P\right)$. Setting up
(34)$$\begin{array}{cc} P\tilde{X}\left(g\right) & =P(\tilde{A}\left(g\right)-\tilde{K}\tilde{C})=P\tilde{A}\left(g\right)-N\tilde{C}, \end{array}$$
(35)$$\begin{array}{c} P\tilde{Z}=P\left({\tilde{W}}_{1}-\tilde{K}{\tilde{W}}_{2}\right)=P{\tilde{W}}_{1}-N{\tilde{W}}_{2}, \end{array}$$
and employing this set up into (Equation 33) leads to (Equation 24), which results in the proof being completed. □

Finally, the design procedure boils down to solving the set of LMIs (Equation 24), and then calculating the gain matrices for the estimator from
(36)$$\tilde{K}=\left[\begin{array}{c} K_{x}\\ K_{s} \end{array}\right]=P^{-1}N.$$

Thus, the arrangement of the entire methodology can be split up into off-line as well as on-line parts and summarized as the following algorithm: **I.**Off-line stage: **Step 1:**Find a feasible solution to the problem (Equation 24) for obtaining ***P*** and ***N***. If there is at least one feasible solution then go to Step 2 else STOP; **Step 2:**Calculate the gain matrices ***K****x* and ***K****s* of the observer with (Equation 36); **Step 3:**Set time k=1. **II.**On-line stage: **Step 1:**Calculate state and fault estimates with (Equation 6)–(Equation 8); **Step 2:**Set k=k+1, and go to Step 1.

The main limitations of the proposed approach are caused by the assumptions, which means that firstly, the uncertainty vectors should belong to the $l_{2}$ class; hence, it means they have a finite energy. The other limitation is that the nonlinear function is to be Lipschitz. It causes that the nonlinear system should satisfy the condition 1 of Lemma 1. The fundamental constraint is that the pair(A,C) and especially $pair(\tilde{A},\tilde{C})$ should be observable.

The objective of the subsequent section is to provide an exemplary results containing the state and sensor fault estimation.

## 4. Exemplary Results

### 4.1. Case Study

For the sake of validation purposes, the proposed algorithm was applied to the MT (multi-tank system) system provided by Inteco Ltd., Kraków, Poland [48] (see Figure 1). It is worth noticing that the system being under the study is fully computer-based controlled, which simplifies the control, identification and estimation strategies being investigated. It is configured in such a way that it consists of three different tanks, which are placed vertically one above the other. It is supplied with a fully controlled flow water pump, which supplies water to the upper tank. The water then flows out to the middle one and finally to the lower one. After that, it flows down to the reservoir. Those tanks are interconnected to each other with fully controlled solenoid valves, and their adjustment is done with a PWM (Pulse Width Modulation) signal. The PWM signal is also used to control the pump. The variables measured in that system are the water levels of the respective tanks. These measurements are based on the water pressure sensors, and the water level in [m] is provided to the user.

During the experiments, the system was operated in a pump-valve mode which relies on controlling the system by both pump and valves, which is the most interesting and the most difficult mode. It allows stabilizing the water level in each tank, and every desired level in each tank is allowed to be achieved. Thus, the control input is given by
(37)$$u_{k}=\left[\begin{array}{c} u_{p,k}\\ u_{v_{1},k}\\ u_{v_{2},k}\\ u_{v_{3},k} \end{array}\right],$$
where $u_{p,k}$ stands for the control signal for the pump, whilst $u_{v_{1},k}$, $u_{v_{2},k}$ and $u_{v_{3},k}$ denote the suitable signals for the drain solenoid valve of the top, middle and lower tanks, respectively, at time *k*. Moreover, the system’s behavior is described by the following state vector $x_{k}={\left[\begin{array}{ccc} :x_{1,k} & x_{2,k} & x_{3,k} \end{array}\right]}^{T}$, where $x_{1,k}$, $x_{2,k}$ and $x_{3,k}$ signify a suitable liquid water of the first, second and third tank, respectively. The distribution matrices of the process and measurement uncertainties were achieved with series of tests. This estimated inaccuracies, and specifically their matrices were established as follows:(38)$$W_{1}=\left[\begin{array}{ccc} 0.41 & 0 & 0\\ 0 & 0.23 & 0\\ 0 & 0 & 0.19 \end{array}\right]\cdot 10^{-4},$$
(39)$$W_{2}=\left[\begin{array}{ccc} 0.4059 & 0 & 0\\ 0 & 0.2277 & 0\\ 0 & 0 & 0.1881 \end{array}\right]\cdot 10^{-2}.$$

The sensor fault distribution matrix *f* is set with components equal to either ones and zeros in which one denotes that the fault acts onto an appropriate sensor reading whilst zero signifies an opposite position. As a consequence, it was set as follows:(40)$$f=\left[\begin{array}{cc} 1 & 0\\ 0 & 1 \end{array}\right].$$

The evaluation was accomplished in such a way that the sensor in the third tank is not taken into account as a possibly faulty one, why is why the level sensor in the bottom tank is considered as never existing, which additionally makes the entire estimation process harder to realize. It entails the output matrix given by
(41)$$C=\left[\begin{array}{ccc} 1 & 0 & 0\\ 0 & 1 & 0 \end{array}\right].$$

It can be summarized that two out of three liquid levels, namely the level in the top tank as well as the level in the middle tank, are measured, and they all are impaired by the faults.

Such a configuration of the system allows defining a scenario for investigating the fault estimation process, which is given as follows:(42)$$\begin{array}{cc} f_{1,k} & =\left\{\begin{array}{cccccc} -0.05 & \; & & \; & & 3000\leq k\leq 5000\\ 0 & \; & & \; & & \mathrm{otherwise} \end{array}\right. \end{array}$$(43)$$\begin{array}{cc} f_{2,k} & =\left\{\begin{array}{cccccc} +0.02 & \; & & \; & & 4000\leq k\leq 5500\\ -y_{2,k} & \; & & \; & & 5500\leq k\leq 7000\\ 0 & \; & & \; & & \mathrm{otherwise} \end{array}\right. \end{array}$$

An examination of the fault estimation approach in four manners is allowed in such a scenario. Firstly, there is a temporary, abrupt fault being biased to the real state. The readings show the value as 5 cm less than it really is in the tank. The second one is the fault in which the measurements in the second tank are impaired by an abrupt fault, in which the values read from the sensor are by 2 cm higher than they really are. Then, the sensor readings suddenly run into stuck in place fault. This means that the readings from this sensor are still very the same irrespective of the actual water level. Specifically, the value read from this sensor is always 0 in the specified period of time. The other aspect is that those two above mentioned faults are partly in the same time span, which additionally might include difficulties during the estimation process. Finally, the fourth aspect is that the sensor in the bottom tank is not taken into account, as it was already mentioned, due to a missing level sensor in the third tank which renders the estimation more difficult in detection.

It is worth noticing that the achieved results by solving the LMI (Equation 24) gave the following gain matrices: (44)$$\begin{array}{cc} K_{x} & =\left[\begin{array}{cc} 0.3749 & 0.0001\\ 0.0002 & 0.3437\\ 0 & 0.0002 \end{array}\right], \end{array}$$(45)$$\begin{array}{cc} K_{s} & =\left[\begin{array}{cc} -0.4293 & 0\\ -0.0002 & -0.4106 \end{array}\right]. \end{array}$$

For further analysis, it should be emphasized that the experiment was carried out in an open loop with control signals set as the ones provided in the Figure 2.

It means that the water pump performed with 50% efficiency throughout the whole time span of the experiment, while the solenoid valves were changing in some sinusoidal ways.

### 4.2. Discussion

Figure 3, Figure 4 and Figure 5 show the response of the system and specifically its particular state ***x***, given by blue solid lines, while red dashed lines stand for the estimation 
$\overset{\wedge }{x}$ signal. The measured output ***y*** are presented in green dash-dotted lines.

It can be easily noticed that the state estimates follow the real states with very high accuracy, despite the quite big noise present in the measurements. The state estimates converged to the real state very quickly in all three cases. Only the third state was biased a bit in the initial phase, but within acceptable limits, although that situation is caused by the fact that this specific state was immeasurable. Taking such a fact into account, the state estimation in that case when it was immeasurable can be perceived as being very proper. In these figures, docked windows show zooms of a specific part of time in which the obtained results can be seen more precisely.

Another important and interesting thing is the sensor fault and especially its estimate. These signals are presented in Figure 6 and Figure 7, where blue dashed lines represent the real faults and red solid lines stand for the estimates of the faults.

It should be emphasized that the real faults are plotted only for demonstration purposes, and their estimates are obtained without any knowledge of their shape and magnitude. They were achieved just on the basis of the model of the system and the structure of the estimator. It can be easily noticed that the sensor faults were estimated with quite good precision. It is obvious that the fault estimates are impaired with some noise, which means that the estimates follow the real ones by oscillating around the specific value with some relatively small amplitude. It might seem that the inaccuracy of the fault estimate of the sensor placed in the top tank is bigger then the one placed in the middle; however, although there is a slight difference, it is not big and could be perceived as a good quality. Although they have been reconstructed precisely regardless of if there was an abrupt fault or stack in place one when the sensor reading was at a constant value, the estimator is capable of reconstructing them appropriately. The docked windows show zooms of a specified time span in which the precision as well as fast convergence can be observed. It can be noticed that the fault estimate very precisely and quickly reacts to changes related to the real faults.

The obtained results for the state as well as sensor faults indicate a very good estimation quality. The achieved results confirm the performance of the proposed approach.

## 5. Conclusions

The paper dealt with the problem of simultaneous state and sensor fault estimation. The investigated problem is rather very common; however, in this paper, a proposed solution in the form of the adaptive observer is slightly different from those presented in literature. It actually combines two approaches, the direct fault estimation which relies on achieving sensor fault estimation directly from the output equation, and the classical adaptive observer. It particularly means that the features of both was received. Firstly, the sensor fault equation was achieved from the output, and based on this, the observer was constructed. The proposed observer also contains the correction part, which additionally stabilizes the estimator contrary to the direct estimator. However, an unwelcome factor of the rate of change of the sensor fault was vanished, comparing to the classical adaptive way of estimating this kind of fault. Moreover, the proposed approach was provided for the class of non-linear systems, and it also can handle the exogenous uncertainties influencing the system. To solve such a problem, a $H_{\infty }$ approach was utilized. The verification of the algorithm was made by implementation to the laboratory multi-tank system. The obtained results clearly confirm the efficiency of the proposed approach. The future works will focus on integrating the proposed approach with the one capable of actuator fault estimation, which will result in the simultaneous estimation of the actuator and sensor faults. Moreover, employing the proposed approach to the FTC scheme is going to be developed. Furthermore, the authors will focus on combining the proposed approach with a suitable ILC scheme.

## Figures and Tables

**Figure 1 sensors-22-09638-f001:**
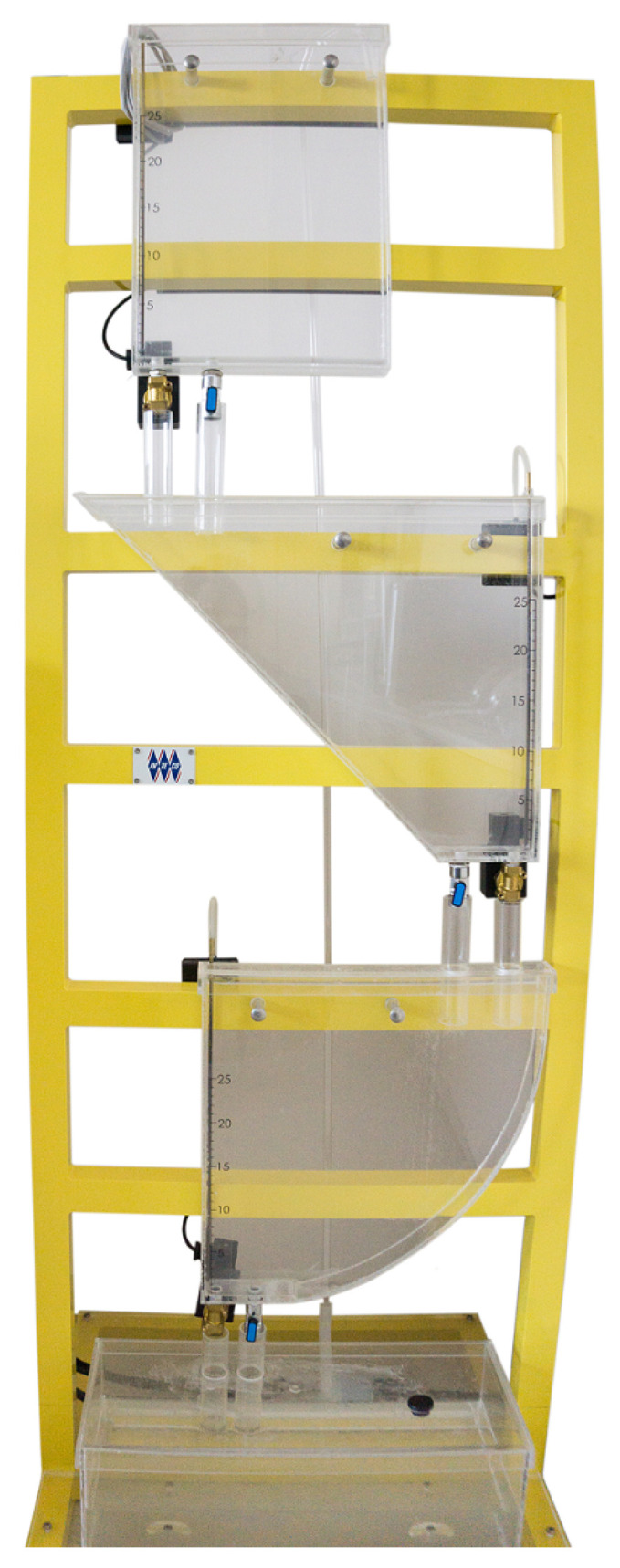
The laboratory multi-tank system.

**Figure 2 sensors-22-09638-f002:**
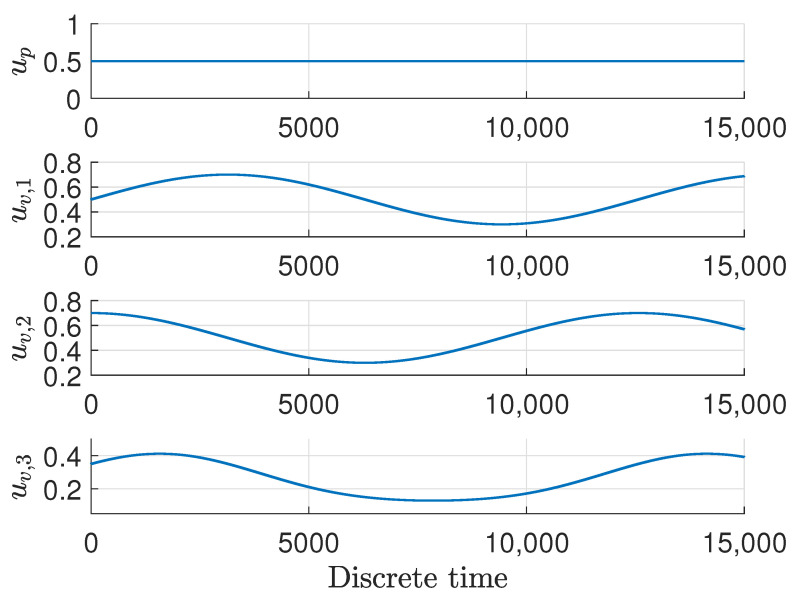
The control signals *u* during the experiment.

**Figure 3 sensors-22-09638-f003:**
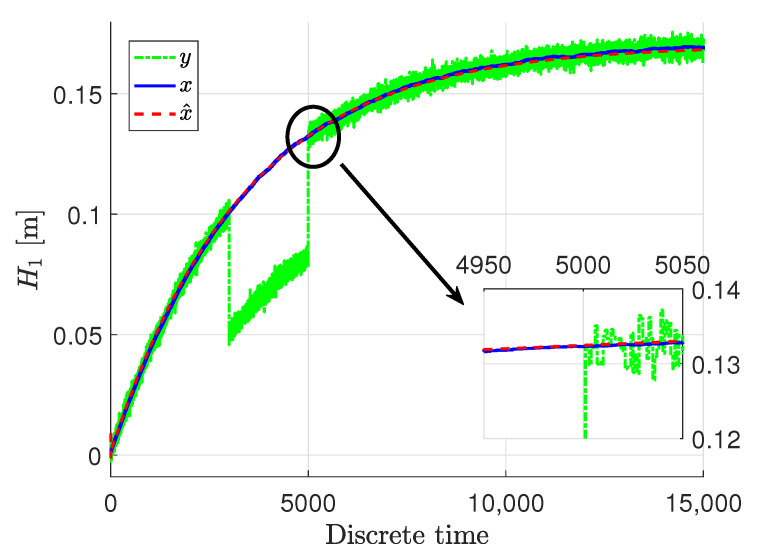
The liquid level $x_{1,k}$ and its estimate ${\overset{\wedge }{x}}_{1,k}$.

**Figure 4 sensors-22-09638-f004:**
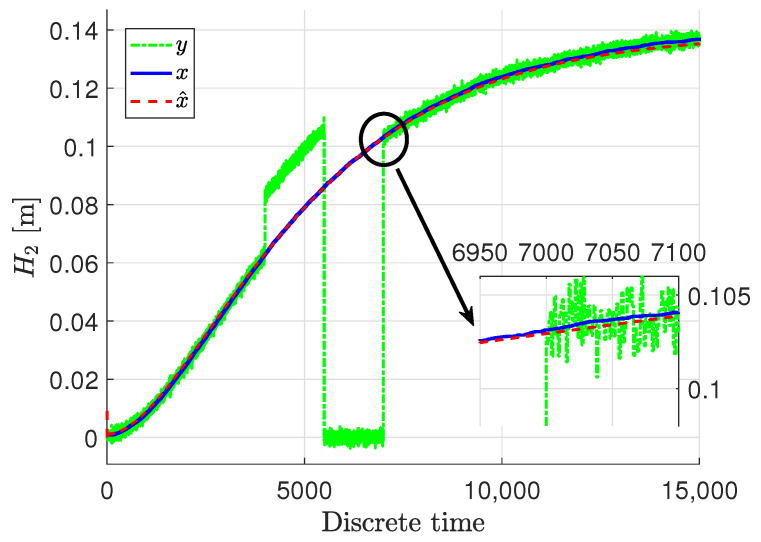
The liquid level $x_{2,k}$ and its estimate ${\overset{\wedge }{x}}_{2,k}$.

**Figure 5 sensors-22-09638-f005:**
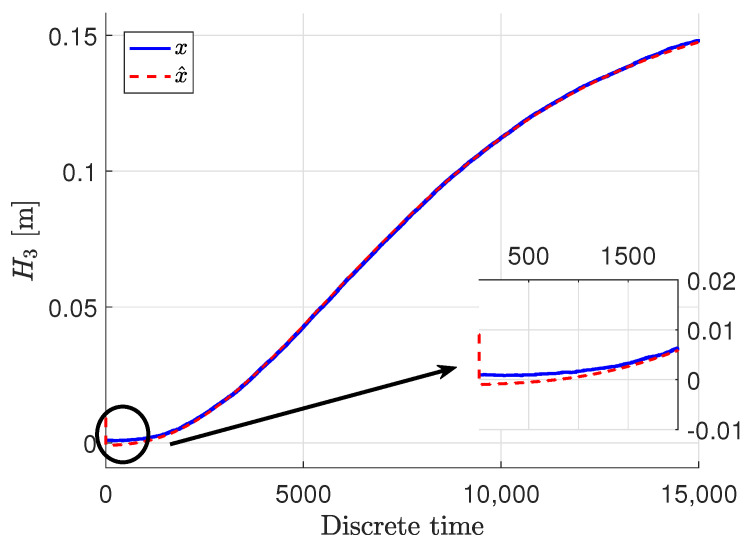
The liquid level $x_{3,k}$ and its estimate ${\overset{\wedge }{x}}_{3,k}$.

**Figure 6 sensors-22-09638-f006:**
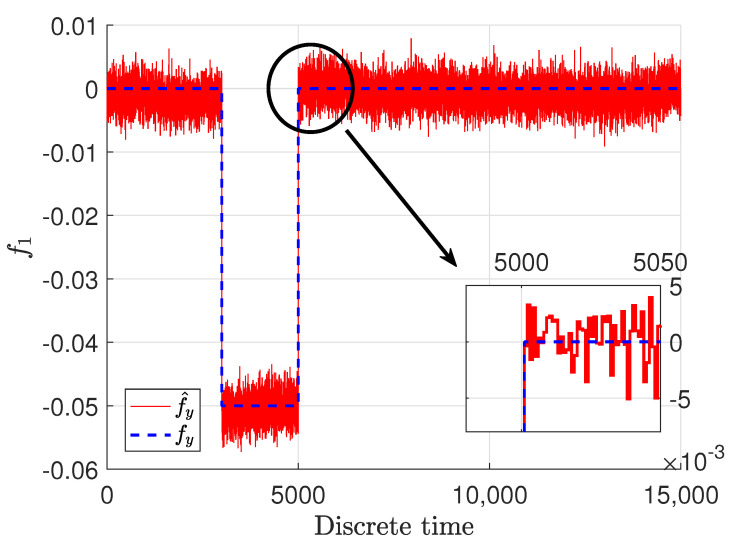
Sensor fault $f_{1,k}$ and its estimate ${\overset{\wedge }{f}}_{1,k}$.

**Figure 7 sensors-22-09638-f007:**
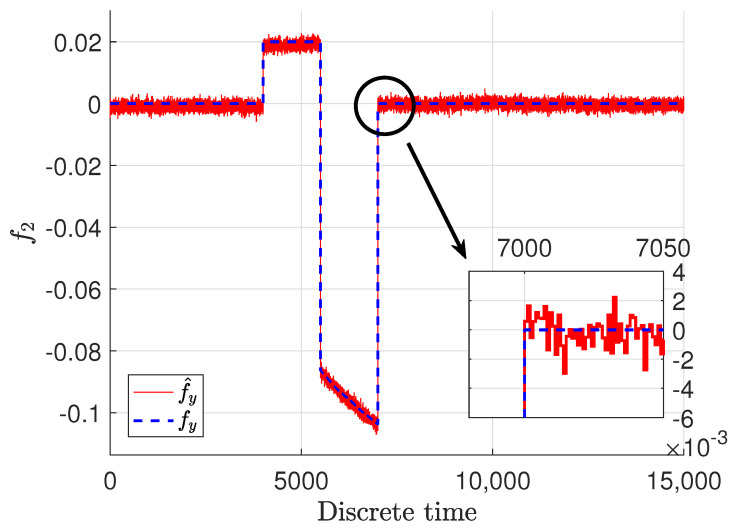
Sensor fault $f_{2,k}$ and its estimate ${\overset{\wedge }{f}}_{2,k}$.

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
