# Peer review of "A Novel Adaptive Sensor Fault Estimation Algorithm in Robust Fault Diagnosis"

_sensors, 2022, doi:10.3390/s22249638_

Round 1

Reviewer 1 Report

1. The language should be improved to increase the readability of this paper.

2. The introduction is too short and there are few references. I think a more comprehensive presentation of the current state of research in this field should be given.

3. The meaning of each symbol in the Equation should be given. Please check all the Equations to make sure the meaning of each symbol is given to increase the intelligibility of this paper.

4. A discussion section should be added to this paper.

5. Limitations and future work should be clarified in this paper.

Author Response

R1-Comment-1. The language should be improved to increase the readability of this paper.

Answer: Thank you very much for this comment. The English language has now been improved by polishing by a professional service of one of the JCR journals while preparing Revision 1 of the review process.

R1-Comment-2. The introduction is too short and there are few references. I think a more comprehensive presentation of the current state of research in this field should be given.

Answer: Thank you for this comment. We totally agree with the reviewer and in the revised version the introduction was suitably extended and more comprehensive presentation of the current state of art is presented.

R1-Comment-3. The meaning of each symbol in the Equation should be given. Please check all the Equations to make sure the meaning of each symbol is given to increase the intelligibility of this paper.

Answer: Thank you very much for this comment. The meaning of each symbol is presented in the table in the revised version just before the “references” section.

R1-Comment-4. A discussion section should be added to this paper.

Answer: Thank you for the comment. A suitable discussion is employed in the revised paper.

R1-Comment-5. Limitations and future work should be clarified in this paper.

Answer: Thank you for such a comment. In the revised paper, the limitations are provided within the main text of the manuscript while the future work is presented in “conclusion” section.

Reviewer 2 Report

The paper deals with a robust sensor fault estimation by proposing a novel algorithm capable for reconstructing faults occuring in the system. Overall, the paper is well written and organized with a proper length. The contributions as well as the quality are both good. In addition, there are some points that are not very clear and should be addressed in the revised version:

1. Please update the references, especially for recent years. For example,  robust fault diagnosis is an important research issue which is widely applied on incipient fault diagnosis, and the following reference had given significant design results:

[1] Incipient winding fault detection and diagnosis for squirrel-cage induction motors equipped
on CRH trains
. ISA Transactions, 2020, 99: 488~495.

2. The Introduction section is too short for a regular paper.
3. The innovation of this paper is not clear and it is difficult for readers to understand the main contributions of this paper.
This part should be added in Introduction section.

 4. The reviewers recommend that more future work should be added on Conclusion Section.

Author Response

R2-Comment-1. Please update the references, especially for recent years. For example,  robust fault diagnosis is an important research issue which is widely applied on incipient fault diagnosis, and the following reference had given significant design results:

[1] Incipient winding fault detection and diagnosis for squirrel-cage induction motors equipped on CRH trains. ISA Transactions, 2020, 99: 488~495.

Answer: Thank you very much for this comment. In the revised version the introduction was suitably extended and more comprehensive presentation of the current state of art is presented which results with the updated references in fault diagnosis, etc.

R2-Comment-2. The Introduction section is too short for a regular paper.

Answer: Thank you very much for this comment. In the revised version the introduction was suitably extended.

R2-Comment-3. The innovation of this paper is not clear and it is difficult for readers to understand the main contributions of this paper. This part should be added in Introduction section.

Answer: Thank you for this comment. The innovation and limitations of the proposed approach has now been added to the manuscript in the revised version.

R2-Comment-4. The reviewers recommend that more future work should be added on Conclusion Section.

Answer: Thank you for such a comment. In the revised paper, the future work is provided in “conclusion” section.

Round 2

Reviewer 1 Report

All my questions have been answered.

Reviewer 2 Report

In a general way most of my comments were answered by the authors. My overall opinion about this paper is quite good. The manuscript is well written and acceptable for publishing